# Identification of Risk Factors Associated with Resistant *Escherichia coli* Isolates from Poultry Farms in the East Coast of Peninsular Malaysia: A Cross Sectional Study

**DOI:** 10.3390/antibiotics10020117

**Published:** 2021-01-26

**Authors:** Sharifo Ali Elmi, David Simons, Linzy Elton, Najmul Haider, Muzamil Mahdi Abdel Hamid, Yassir Adam Shuaib, Mohd Azam Khan, Iekhsan Othman, Richard Kock, Abdinasir Yusuf Osman

**Affiliations:** 1Faculty of Veterinary Medicine, Universiti Malaysia Kelantan, Pengkalan Chepa, Kota Bharu 16100, Kelantan, Malaysia; sharifo.d18d006f@siswa.umk.edu.my (S.A.E.); azamkhan@umk.edu.my (M.A.K.); 2The Royal Veterinary College, University of London, Hawkshead Lane, North Mymms, Hatfield AL9 7TA, Hertfordshire, UK; dsimons19@rvc.ac.uk (D.S.); nhaider@rvc.ac.uk (N.H.); rkock@rvc.ac.uk (R.K.); 3Centre for Clinical Microbiology, University College London, London NW3 2PF, UK; linzy.elton@ucl.ac.uk; 4Institute of Endemic Diseases, University of Khartoum, Khartoum 11111, Sudan; mahdi@iend.org; 5College of Veterinary Medicine, Sudan University of Science and Technology, Hilat Kuku, Khartoum North 13321, Sudan; vet.aboamar@gmail.com; 6Jeffrey Cheah School of Medicine and Health Sciences, Monash University, Selangor 47500, Malaysia; iekhsan.othman@monash.edu

**Keywords:** antimicrobial resistant, *Escherichia coli*, distribution, poultry farms, environment, east coast of peninsular Malaysia

## Abstract

Antimicrobial resistance is of concern to global health security worldwide. We aimed to identify the prevalence, resistance patterns, and risk factors associated with *Escherichia coli* (*E. coli*) resistance from poultry farms in Kelantan, Terengganu, and Pahang states of east coast peninsular Malaysia. Between 8 February 2019 and 23 February 2020, a total of 371 samples (cloacal swabs = 259; faecal = 84; Sewage = 14, Tap water = 14) were collected. Characteristics of the sampled farms including management type, biosecurity, and history of disease were obtained using semi-structured questionnaire. Presumptive *E. coli* isolates were identified based on colony morphology with subsequent biochemical and PCR confirmation. Susceptibility of isolates was tested against a panel of 12 antimicrobials and interpreted alongside risk factor data obtained from the surveys. We isolated 717 *E. coli* samples from poultry and environmental samples. Our findings revealed that cloacal (17.8%, 46/259), faecal (22.6%, 19/84), sewage (14.3%, 2/14) and tap water (7.1%, 1/14) were significantly (*p* < 0.003) resistant to at least three classes of antimicrobials. Resistance to tetracycline class were predominantly observed in faecal samples (69%, 58/84), followed by cloacal (64.1%, 166/259), sewage (35.7%, 5/14), and tap water (7.1%, 1/84), respectively. Sewage water (OR = 7.22, 95% CI = 0.95–151.21) had significant association with antimicrobial resistance (AMR) acquisition. Multivariate regression analysis identified that the risk factors including sewage samples (OR = 7.43, 95% CI = 0.96–156.87) and farm size are leading drivers of *E. coli* antimicrobial resistance in the participating states of east coast peninsular Malaysia. We observed that the resistance patterns of *E. coli* isolates against 12 panel antimicrobials are generally similar in all selected states of east coast peninsular Malaysia. The highest prevalence of resistance was recorded in tetracycline (91.2%), oxytetracycline (89.1%), sulfamethoxazole/trimethoprim (73.1%), doxycycline (63%), and sulfamethoxazole (63%). A close association between different risk factors and the high prevalence of antimicrobial-resistant *E. coli* strains reflects increased exposure to resistant bacteria and suggests a concern over rising misuse of veterinary antimicrobials that may contribute to the future threat of emergence of multidrug-resistant pathogen isolates. Public health interventions to limit antimicrobial resistance need to be tailored to local poultry farm practices that affect bacterial transmission.

## 1. Introduction

Antimicrobial resistance (AMR) is of concern to global health security [1]. The persistence and emergence of antimicrobial resistance in bacterial communities with special reference to faecal form indicators pose a threat to treatment options of microbial infections, and thus place a burden on health services in human and animal health settings [2]. Moreover, the production of poultry for food relies on the use of antimicrobials to ensure animal health and growth promotion under intensive farming conditions [3]. Most of these antimicrobial compounds used in poultry operations are accumulated and biomagnified through the food chain. Exposure among local human populations to low levels of antimicrobial environmental contaminants through marine and agricultural ecosystems has been proposed to lead to development and acquisition of resistant bacteria [4,5,6].

*Escherichia coli* (*E. coli*) is an important pollution indicator with pathogenic strains responsible for food poisoning and food related infections. In upper middle-income countries, *E. coli* is responsible for 25% of infant diarrhea and some enteropathogenic, enter invasive, and enterotoxigenic types of *E. coli* are leading causes of food-borne diarrhea [7]. The prevalence of AMR among food-borne pathogens has increased during recent decades [7,8]. Factors influencing bacterial resistance on farms are substantial, including flock health status, farm management practices, and the environment [9]. Practices such as rampant use of broad-spectrum antimicrobials administered in low doses for growth promotion [10,11,12,13] and use of non-approved drugs or drugs used in off-label scenarios are driving the emergence of antimicrobial resistance in veterinary settings [14]. The tangled interplay of antimicrobial use and microbial transmission between people, animals, and the environment complicates efforts to reduce the development of AMR.

In Malaysia, these issues are likely to be most acute in poultry operations. Use of antimicrobials is frequently coupled with a high prevalence of infectious disease [15]. Malaysia has one of the largest poultry industries in South East Asia. Rapid growth and intensification in the production of chickens for food has the potential to increase the risk of development of AMR strains of pathogenic bacteria. In particular, the east coast of Malaysia has experienced rapid technological, genetic, management, and structural changes within the poultry production industry. In the absence of coordinated and systematically implemented regulation, AMR has been consistently reported at high levels from the poultry industry. However, little is known about the risk factors of AMR in poultry operations in peninsular Malaysia. Comprehensive examinations of a range of ultimate and proximate drivers of AMR at the poultry farm level have not been thoroughly investigated in South East Asia. To better understand the interactions of factors driving AMR across smallholders and poultry operations in Malaysia, we aimed to identify risk factors associated with the carriage of resistant *E. coli* isolates to help inform antimicrobial stewardship policy in poultry farms in Malaysia.

## 2. Results

We administered a semi-structured questionnaire to 31 poultry farmers and sampled only conveniently 14 farms with a total of 371 samples across three states of peninsular Malaysia. The socio-demographic traits of poultry farmers participating in the surveys are given in Appendix A. Of these 371 samples from 14 poultry farms, the following types were collected: cloacal swabs = 259; faecal = 84; Sewage = 14, tap water = 14. A total of 717 *E. coli* samples were isolated from poultry and environmental samples, as follows: (72%, 519/717) in cloacal swab; (24%, 172/717) in faecal; 20 (2.8%, 20/717) in tap water; and 6 (0.83%, 6/717) in sewage system. A summary of the prevalence of tested *E. coli* samples were given in Table 1. The prevalence of *E. coli* among Kelantanese farms (72.8%, 115/158) was higher than those of Terengganu (57.5%), 46/80) and Pahang (57.9%, 77/113) (Table 1). Among the districts, the highest prevalence of *E. coli* was recorded in Jeli farms (88.5%, 23/26) followed by Machang (85.7%, 24/28) and Kuala Terengganu (76.9%, 20/26), respectively (Table 1). Similarly, the prevalence of *E. coli* was higher in cloacal (66.4%, 259/172) and faecal samples (69%, 58/84)) than sewage (35.7%, 5/14) and tap water (21.4%, 3/14) (Table 1). A high prevalence of resistance to common antimicrobials was observed with special reference to tetracycline (91.4%), oxytetracycline (88.4%), sulfamethoxazole/trimethoprim (74.2%), doxycycline (66.4%), and sulfamethoxazole (65.5%), ampicillin (51.9%), and nalidixic acid (52.2%), but there is a low resistance to chloramphenicol (26.3%), gentamicin (23.3%), amoxicillin (21.2%), ciprofloxacin (19.4%), and cefoxitin (6.5%) (Figure 1). We observed that the resistance patterns of *E. coli* isolates against 12 panel antimicrobials are generally similar in all selected states of east coast peninsular Malaysia that include Kelantan, Terengganu, and Pahang. However, the prevalence of resistance to tetracycline, oxytetracycline, sulfamethoxazole/Trimethoprim, sulfamethoxazole, and doxycycline was consistently higher than other tested antimicrobials across selected states of east coast peninsular Malaysia (Figure 2). The percentage of antimicrobial-resistant *E. coli* isolated from each sample are summarized in Figure 3. Cloacal and faecal samples had the highest percentage of resistance followed by sewage and tap water systems (Figure 3). The summary of resistance to at least one antimicrobial and their associated risk factors is shown in Table 2. We observed that the size of the farm (*p* < 0.023), and source of the water (*p* < 0.02), poultry origin (*p* < 0.01), and the source of the sample (*p* < 0.01) factors were significantly associated with at least one AMR (Table 2). Furthermore, our findings revealed that cloacal (17.8%, 46/259), faecal (22.6%, 19/84), sewage (14.3%, 2/14), and tap water ((7.1%, 1/14) were significantly (*p* < 0.003) associated with resistant to at least three classes of antimicrobial (Table 3). Resistance to tetracycline class were predominantly observed in faecal samples (69%, 58/84), followed by cloacal (64.1%, 166/259), sewage (35.7%, 5/14), and tap water (7.1%, 1/84), respectively (Table 3). Similarly, resistance to quinolones class was predominantly recorded in cloacal samples (45.2%, 117/259), followed by faecal (41.7%, 35/84), sewage (7.1%, 1/14) and tap water (7.1%, 1/14), respectively (Table 3). Sewage water (OR = 7.22, 95% CI = 0.95–151.21) had an increased likelihood of AMR acquisition (Table 4).

Bacteria in samples obtained from young chickens (OR = 1.2, 95% CI = 0.79–1.84) had an increased likelihood of AMR compared to samples from adult chickens (Table 4). Of note, in unadjusted analysis, there was no important difference in the odds of sampled *E. coli* having identified in AMR between intensive, mixed OR = 1.11, 95% CI = 0.49–2.66, or semi-intensive farms. Similarly, no difference in unadjusted analysis was observed in the production system. The results of the multivariate regression analysis adjusting for the size of the farm identified that the risk factors include the source of samples with special reference sewage samples (OR = 7.43, 95% CI = 0.96–156.87) and farm size (small = OR = 2.50, 95% CI = 1.33–4.77; medium = OR = 1.55, 95% CI = 0.89–2.67) as leading drivers of *E. coli* antimicrobial resistance in the participating states of east coast peninsular Malaysia (Table 5).

For PCR analysis, the resistance genes, *aac (3)-IV* for gentamicin, *tet (A)* and *tet (B)* for tetracyclines, *catA1* for chloramphenicol, and *sul1* for sulfonamides were investigated and the proportion of positive resistance genes were given in Table 6. Our results revealed 100% positive amplicons for the *sul1* gene, followed by *aac (3)-IV* 87%, 64.2% of the *E. coli* isolates carried *tet (A)* and *tet (B)* (Table 6). The set of primers used for each gene is given in Table 7.

## 3. Discussion

In this study, we observed that the resistance patterns of *E. coli* isolates against 12 panel antimicrobials are generally similar in all selected states of east coast peninsular Malaysia that include Kelantan, Terengganu, and Pahang. However, there is substantial heterogeneity in the prevalence of *E. coli* AMR within and between these states. These differences in prevalence across these states are linked to geographic-specific risk factors. The prevalence of *E. coli* resistance to tetracycline, oxytetracycline, sulfamethoxazole/trimethoprim, doxycycline, and sulfamethoxazole was highly consistent in all three participating states. This resistance also reflects the common use of antimicrobials in these poultry operations as well as in other agricultural activities [21]. Moreover, most of these antimicrobials are also used in human medicine with special reference to tetracycline, oxytetracycline, sulfamethoxazole, and ampicillin. Our findings are similar to those of other studies in poultry farms in low-income settings in South East Asia (SEA) [22]. For instance, poultry sampling farms in Vietnam found similar proportions of *E. coli* resistant to ampicillin (86.0%), tetracycline (93.4%) oxytetracycline (93·6%), trimethoprim/sulfamethoxazole (69.7%), nalidixic acid (80.1%), gentamicin (19.9%), and chloramphenicol (51·5%). These farm-level estimates are based on non-randomly selected samples and we should expect these estimates to be higher than estimates from random collected datasets [22,23]. For example, *E. coli* isolated from poultry specimens presented at a veterinary clinic in the northern region of peninsular Malaysia were highly resistant to ampicillin (92.7%), tetracycline (91.6%), doxycycline (86.4), and gentamicin (41.6) [24]. Implementation of biosecurity levels including sewage system, visitors, PPE, washing facilities, use of disinfectant, and source of the food were not important factors of *E. coli* antimicrobial resistant in the sampled poultry farms. Furthermore, the prevalence of *E. coli* resistance in cloacal, faecal, sewage, and tap water isolates were significantly (*p* < 0.003) associated with AMR acquisition. Importantly, sewage isolates (OR = 7.43, 95% CI = 0.96–156.8) had an increased testing of AMR as the sewage systems nearby these farms were identified as important risk factors for the presence of AMR. The resistance data from sewage samples can be augmented well with data from clinical based surveillance [25]. The lower prevalence in sewage and tap water isolates, however, could be correlated with sensitivity as it is likely lower than isolate-based surveillance [26]. Resistance to tetracycline class were predominantly observed in faecal isolates, followed by cloacal, sewage, and tap water, respectively. Similarly, resistance to quinolones class were predominantly recorded in cloacal isolates, followed by faecal, sewage, and tap water, respectively.

The source of water and the presence of a sewage system were identified as important risk factors for the presence of AMR in *E. coli* isolates in the study sites. For instance, the pump water OR = 2.02, 95% CI = 1.22–3.36, *p* < 0.000 and surface water OR = 1.57, 95% CI = 0.93–2.68, *p* < 0.000 was significantly associated with AMR acquisition. Furthermore, we detected residual amounts of the antimicrobials tested in the water systems of these premises and alongside the tributaries in the nearby rivers, which is in close proximity to livestock operations (data not presented here). Most of these antimicrobial residues belong to Sulfonamides and Quinolones in the surface water at an average concentration range of 5 to 85 ng L^−1^. Conversely, we have detected low levels of Tetracyclines in the surface water, although higher levels of TCs were detected in the faecal samples of poultry operations and in the Kelantanese tributaries sediments. The discovery of these antimicrobial contents could be most likely attributed to potential contaminations from livestock farming discharge field runoff. Importantly, the sampled poultry farms usually access drinking water from intact sources, and thus the association could reflect contact transmission at the farm level. This association has important implications for low-income countries, where potable water remains a pressing challenge [27]. Consumption of poultry meat and its products is increasing, and most poultry meat and eggs are produced and distributed through informal sources that operate outside national quality-control standards and regulations [28].

Nonetheless, it is worth noting that our study is comparable within the local context of East coast peninsular Malaysia but that there are limited studies conducted in these areas [29,30]. Pathogen transmission could lead to rampant use of veterinary antimicrobials by these farmers as our self-reported data did show explicitly such association. In our findings, we have observed the association between antimicrobial-resistant *E. coli* and the type of production system, although such phenomena were not consistent across all tested antimicrobials. Importantly, although there was a link between washing facilities and antimicrobial-resistant bacteria, such an association, however, was not an important factor for pathogen transmission dynamics. Interestingly, small scale poultry farms in the selected states were far more likely to carry AMR-resistant *E. coli* (OR = 2.33, 95% CI = 1.27–4.35) than medium and large scales farms. The poultry farms practicing intensive management system and the samples obtained from young chickens had increased odds of testing positive for antimicrobial resistant *E. coli*. Our findings highlight that the current strategies to tackle global antimicrobial resistance should include identifying the persistence and drivers of antimicrobial resistance within the context of cultural and management practices in the relevant communities.

Use of antimicrobials was very high in our survey (100%) and 64.5% of participants reported that they had received them from a regulated drug supplier (Appendix A). This suggests that many small holders may buy unregulated medicine from black markets and thus contributes to the development of AMR. Furthermore, it also reflects the national need for a policy to regulate the safety of antimicrobials and guidance for usage and sale. A “One Health” approach involving different actors such as human and veterinary medicine, agriculture, finance, environment, and consumers will be a utopian model to combat global AMR.

In the current study, the *sul1* gene was detected in 100% using the conventional PCR from the poultry samples of east coast peninsular Malaysia. Similarly, the *aac (3)-IV* was detected in 87% where 64.2% of the *E. coli* isolates carried *tet (A)* and *tet (B)*. However, it is worth noting that there are no comparable existing studies which have investigated the presence of these genes in poultry operations of east coast peninsular Malaysia. The detection of resistant *E. coli* genes in rural surface water which is in close proximity to poultry operations remains a source of concern and suggests a potential pool of veterinary antimicrobials and resistant bacteria to the community. This study demonstrated the rampant use of veterinary antimicrobials in poultry operations, which is probably responsible for AMR in community settings. A close association between different risk factors and the high prevalence of antimicrobial-resistant *E. coli* strains reflects the increased exposure to resistant bacteria and suggests a concern over rising misuse of veterinary antimicrobials that may result in a future threat of emergence of multidrug-resistant pathogen isolates. Public health interventions to limit antimicrobial resistance need to be tailored to local poultry farm practices that affect bacterial transmission.

## 4. Materials and Methods

### 4.1. Study Area

The study was carried out between 8 February 2019 and 23 February 2020 in poultry farms located in three states of East coast peninsular Malaysia: Kelantan, Terengganu, and Pahang (Figure 4). These three states border the South China Sea and are dominated by a tropical climate which is characterized by humidity. There is a heavy monsoon season from November to March every year. The average temperature ranges from 21 to 32 °C. Average yearly rainfall falls is from 2032 mm to 2540 mm, with the wettest months being from November through January.

### 4.2. Study Design, Definitions, and Data Sources

We conducted a cross sectional survey targeting poultry farms in three states of east coast peninsular Malaysia that include Kelantan, Terengganu, and Pahang. A total of 371 samples (cloacal swabs = 259; faecal = 84; Sewage = 14, tap water = 14) were randomly collected. Farm characteristics including management, biosecurity, and disease history were collected using a semi-structured questionnaire. As such, 31 farmers that met strict inclusion criteria of keeping poultry farms and who responded to written consent were included in the analyses. Data pertaining to potential risk factors including management, biosecurity, and disease history were collected using semi-structured questionnaires. Similarly, antimicrobial usage data was obtained using a count-based approach, representing the use (yes/no) of an antimicrobial at the time of visit. Furthermore, sources from where antimicrobials and feed along with the source of water and the nature of their current sewage systems were collected (Appendix A). 

Regarding the management system, flock size, and sewage system, the following definitions and criteria were used:Intensive management system is defined as mainly concentrated and often mechanized operations that use controlled-environment systems to provide the ideal thermal environment for the poultry.Semi-intensive system is that which relies on natural airflow though the shed for ventilation.Extensive system is mainly pasture-based and land-based where birds in the household flock are typically housed overnight in the shelter and are let out in the morning to forage during the day.The criteria of the farm size included large-scale commercial farms that has more than ≥10,000 birds, medium-scale commercial farms that has more 5000–10,000, and small-scale farms where birds are often kept in single-age groups of >1000.A poor sewage system is defined as that which retains high volumes of wastewater with low flow rate, blackish appearance, and sewage smell odour as a result of composing agricultural waste—probably as leakage from nearby irrigated effluent which is used for agricultural land application along with the presence of food waste, green waste, plastic, and heavy materials.A good sewage system is that which has good drainage with no agricultural waste and relatively low heavy materials.Excellent swage system is that which has significant drainage, no agriculture, and heavy materials.

Briefly, the cloacal samples were collected using sterile transport media; faecal samples using sterile containers and water samples using sterile water bottles and kept in a cooling box containing ice bags maintaining low temperature at (4°) and transferred to the lab within 24 h. All samples were collected according to standard operating procedures and good laboratory practices. A detailed study design is summarized in Figure 5.

### 4.3. Microbiological Testing

All cloacal swabs and fresh faecal samples were placed in Amies transport media, and transported on ice to the molecular biology laboratory, Universiti Malaysia Kelantan (UMK). Sewage tap water and surface water samples were transported in conical tubes, all on ice. Samples were enriched in buffered peptone water for 24 h and then plated onto eosin methylene blue agar (EMBA) and incubated for 24 h at 37 °C. Subsequently, five colonies were selected and sub-cultured on EMBA, before being further sub-cultured on Müller-Hinton agar and stored at −20 °C in cryovials. A single colony was picked at random from the plate for each original sample and biochemical tests including triple sugar iron agar, Simmon’s citrate agar, and motility-indole-lysine media were used for presumptive identification of *E. coli* isolates. All isolates were revived and inoculated onto Müller-Hinton plates before antimicrobial susceptibility testing.

### 4.4. Antimicrobial Susceptibility Testing

Isolates were tested for susceptibility against a panel of 12 antimicrobial agents perceived to be used frequently in both human and veterinary medicine in Malaysia. These antimicrobials included ampicillin (10 µg), amoxicillin-clavulanic acid (20/10 µg), chloramphenicol (30 µg), gentamicin (10 µg), tetracycline (30 µg), Oxytetracycline (30µg), doxycycline (30µg), trimethoprim-sulfamethoxazole (25 µg), nalidixic acid (30 µg), ciprofloxacin (5 µg), cefoxitin (30µg), and sulfonamides (300 µg) using the disc diffusion method (DDM) according to the Clinical and Laboratory Standards Institute guidelines [31]. Clinical and Laboratory Standards Institute guidelines were also used to determine as breakpoints for classifying isolates as sensitive, intermediate, or resistant to the drug [31]. Multidrug-resistant *E. coli* was defined as “non-susceptibility to at least one agent in three or more antimicrobial classes.” An antibiogram was defined as the combination of antimicrobials to which an isolate was resistant, and thus antibiogram length was defined as the number of antimicrobials to which an isolate was phenotypically resistance.

### 4.5. PCR

#### 4.5.1. DNA Extraction of Escherichia Coli Isolates

*E. coli* isolates were sub-cultured overnight in Luria-Bertani broth and genomic DNA was extracted using a Presto™ Mini gDNA Bacteria Kit according to the manufacturer’s instructions.

#### 4.5.2. Primers and PCR Assay for Specific Genes

The incidence of genes related to resistance to gentamicin (*aac (3)-IV*), tetracyclines (*tet (A) and tet (B)*), chloramphenicol (*catA1 and cmlA*), and sulfonamides (*sul1)* was determined by PCR. The set of primers used for each gene is shown in Table 6. PCR reactions were performed in a total volume of 25 mL using GoTaq1 Green Master Mix (Promega, USA), including 12.5 mL of GoTaq1 Green Master Mix, 1 mL of forward primer, 1 mL of reverse primers, 5.5 mL of nuclease-free water, and 5 mL of extracted DNA. Amplification reactions were carried out using a DNA thermocycler (Fisher Scientific UK, Loughborough, UK). PCR amplification was performed in duplicate. Amplified samples were analysed by electrophoresis in 1.5% agarose gel and were stained with ethidium bromide.

### 4.6. Statistical Analysis

Data were entered into Microsoft Excel spreadsheet and imported into SPSS version 25 and the R software (version 3.6.1) for statistical analysis. The data were sorted and checked for consistency and duplication. Data visualization were done in ArcGIS v. 10 (esri Inc., Redlands, CA, USA). The data focused on sets of variables that have been previously proposed or identified as risk factors for antimicrobial resistance [32,33]. Briefly, we have classified resistance as no resistance to antimicrobials detected in isolates and categorized the antimicrobials into their classes then identified which isolates were resistant to one or more specific classes. Classes of antimicrobials included tetracyclines, penicillins, aminoglycosides, quinolones, sulfonamides, third generation cephalosporins, and chloramphenicol. Prevalence of resistance of *E.coli* to a panel of 12 antimicrobials was also compared between four epidemiological samples that include cloaca, faecal, tap water, and sewage samples. Descriptive statistics for frequency of association between AMR and potential risk factors was performed. Selection of variables for inclusion in a logistic regression model were based on prior hypotheses and variables which were suggestive of an important effect from the descriptive analysis.

## Figures and Tables

**Figure 1 antibiotics-10-00117-f001:**
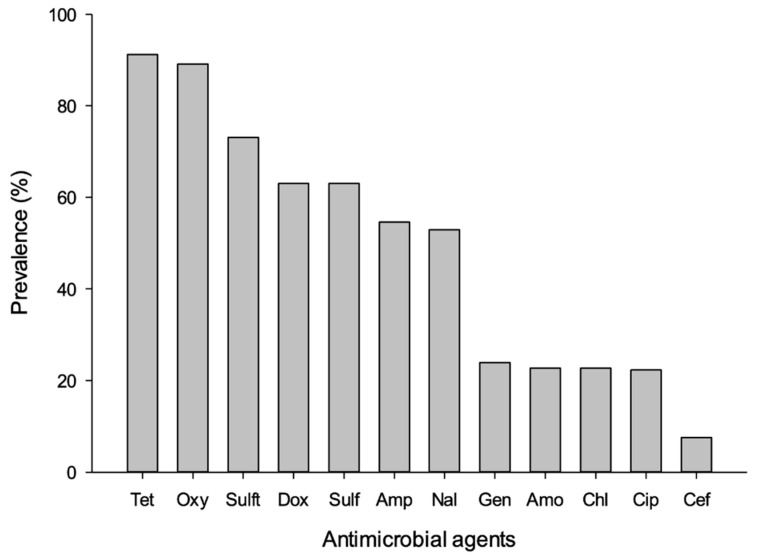
Prevalence of antimicrobial-resistant *Escherichia coli* isolated from poultry farms collected from Kelantan, Terengganu, and Pahang poultry operations. Data are the number of samples (n = 371). Tet: Tetracycline; Oxy: Oxytetracycline; Sulft: Sulfamethoxazole/trimethoprim; Sul: Sulfamethoxazole; Dox: Doxycycline; Amp: Ampicillin; Nal: Nalidixic acid; Chl: chloramphenicol; Gen: Gentamycin; Cip: Ciprofloxacin; Amo: amoxicillin and Cef: cefoxitin.

**Figure 2 antibiotics-10-00117-f002:**
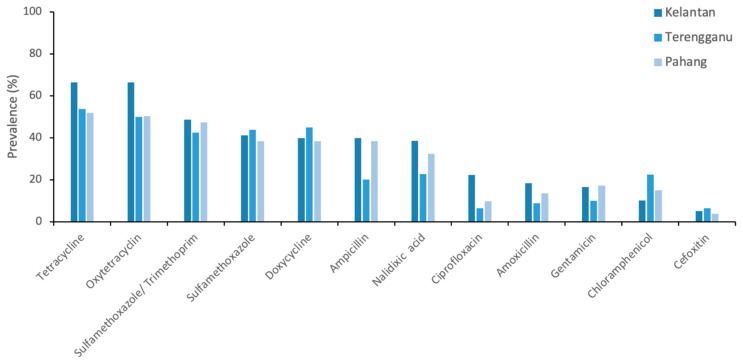
Prevalence of antimicrobial-resistant *Escherichia coli* isolated from poultry farms collected from Kelantan, Terengganu, and Pahang poultry operations. Data are the number of samples (n = 371).

**Figure 3 antibiotics-10-00117-f003:**
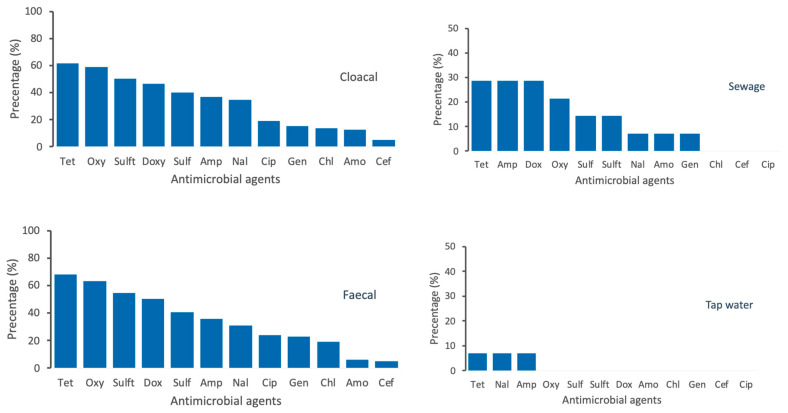
Percentage of antimicrobial-resistant *Escherichia coli* isolated from four epidemiological samples that include cloacal, faecal, tap water, and sewage collected from poultry farms. Data are the number of poultry samples (n = 371) in three states of east coast peninsular Malaysia. Tet: Tetracycline; Oxy: Oxytetracycline; Sulft: Sulfamethoxazole/trimethoprim; Sul: Sulfamethoxazole; Dox: Doxycycline; Amp: Ampicillin; Nal: Nalidixic acid; Chl: chloramphenicol; Gen: Gentamycin; Cip: Ciprofloxacin; Amo: amoxicillin and Cef: cefoxitin.

**Figure 4 antibiotics-10-00117-f004:**
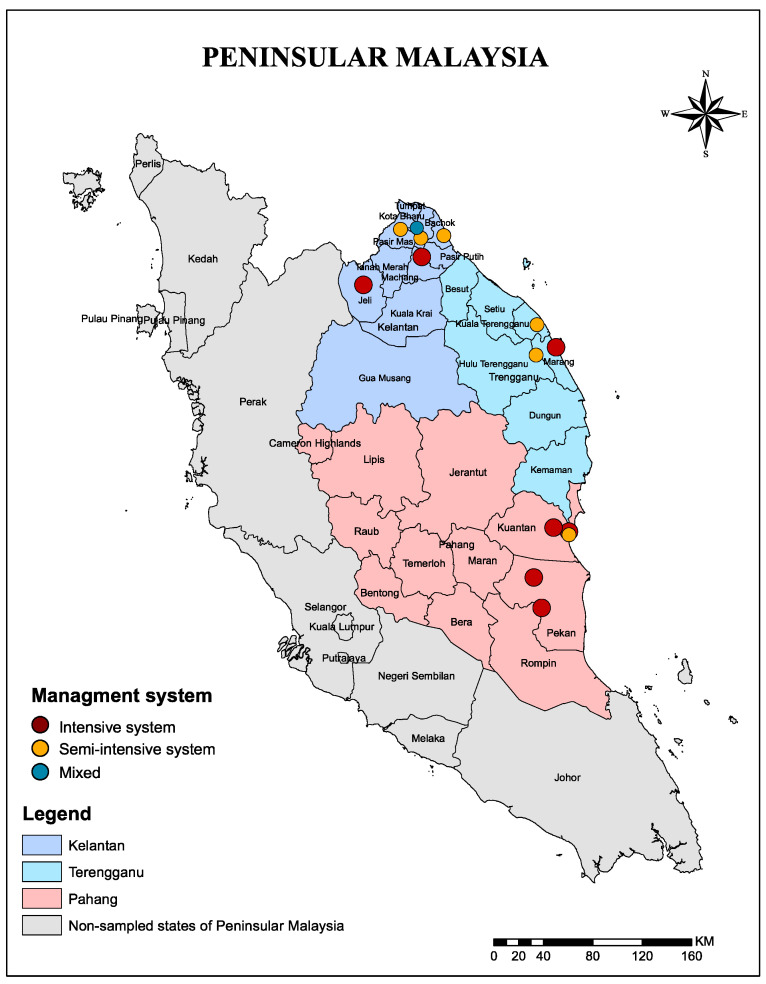
A map showing location of the sampled states and exact location 14 poultry farms sampled and their management systems in Kelantan, Terengganu, and Pahang of east cast peninsular Malaysia. The map was created using ArcGIS v. 10 (esri Inc., Redlands, CA, USA).

**Figure 5 antibiotics-10-00117-f005:**
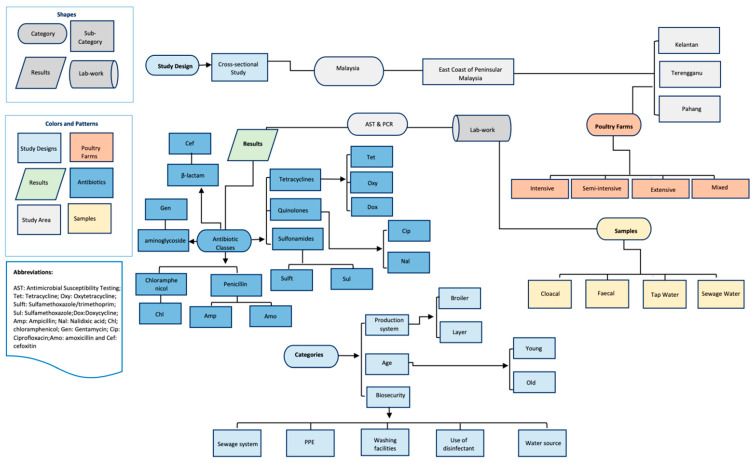
Diagrammatic representation of the study design.

**Table 1 antibiotics-10-00117-t001:** Summary of risk factors of *E.coli* among poultry farms in the Kelantan, Terengganu and Pahang, Malaysia (n = 371) by using chi-square test.

Risk Factors	Samples Tested	Affected (%)	*p*-Value
Age			0.511
Young	187	123 (65.8%)	
Adult	184	115 (62.5%)	
Management system			0.541
Intensive	187	115 (61.5%)	
Semi-intensive	158	105 (66.5%)	
Mixed	26	18 (69.2%)	
Production system			0.278
Broiler	212	129 (60.8%)	
Layer	53	35 (66%)	
Mixed	106	74 (69.8%)	
State			0.012
Kelantan	158	115 (72.8%)	
Terengganu	80	46 (57.5%)	
Pahang	133	77 (57.9%)	
Districts			0.001
Bachok	52	36 (69.2%)	
Kota Bharu	26	18 (69.2%)	
Machang	28	24 (85.7%)	
Pasir Mas	26	14 (53.8%)	
Jeli	26	23 (88.5%)	
Kuantan	79	49 (62%)	
Pekan	54	28 (51.9%)	
kuala terengganu	26	20 (76.9%)	
Marang	54	26 (48.1%)	
Sample source			0.001
Cloaca swab	259	172 (66.4%)	
Faecal Sample	84	58 (69%)	
Sewage	14	5 (35.7%)	
Tape Water	14	3 (21.4%)	
Farm size			0.013
Small	104	77 (74%)	
Medium	188	119 (63.2%)	
Large	79	42 (53.2%)	
Origin of the poultry			0.005
Local	26	18 (69.2%)	
Imported	133	71 (53.4%)	
Both	212	149 (70.3%)	

**Table 2 antibiotics-10-00117-t002:** Summary of prevalence of resistance to at least one antimicrobial and their associated risk factors.

Risk Factors	No Antimicrobial Resistance n = 137	Resistance to at least One Antimicrobial n = 234	*p*-Value
Age			0.44
Young	65 (47.4%)	122 (52.1%)	
Adult	72 (52.6%)	112 (47.9%)	
Origin of the poultry			0.01
Local	10 (7.3%)	16 (6.8%)	
Imported	63 (46%)	70 (29.9%)	
Both	64 (46.7%)	148 (63.2%)	
Management system			0.18
Intensive	80 (58.4%)	115 (49.1%)	
Semi-intensive	47 (34.3%)	103 (44%)	
Mixed	10 (7.3%)	16 (6.8%)	
Production system			0.21
Broiler	86 (62.8%)	125 (53.4%)	
Layer	18 (13.1%)	37 (15.8%)	
Mixed	33 (24.1%)	72 (30.8%)	
Farm size			0.02
Small	29 (21.2%)	75 (32.1%)	
Medium	70 (51.1%)	117 (50%)	
Large	38 (27.7%)	42 (17.9%)	
Source of sample			<0.001
Cloacal swab	89 (65%)	170 (72.6%)	
Faecal sample	26 (19%)	58 (24.8%)	
sewage	9 (6.6%)	5 (2.1%)	
Tap water	13 (9.5%)	1 (0.4%)	
Water source			0.02
Surface water	37 (27%)	69 (29.5%)	
Bond water	61 (44.5%)	72 (30.8%)	
Pump water	39 (28.5%)	93 (39.7%)	
Sewage system			0.60
Excellent	38 (27.7%)	71 (30.3%)	
Good	82 (59.9%)	128 (54.7%)	
Poor	17 (12.4%)	35 (15%)	
Feed source			0.53
Endogenous	50 (36.5%)	82 (35%)	
Exogenous	75 (54.7%)	138 (59%)	
Other	12 (8.8%)	14 (6%)	

**Table 3 antibiotics-10-00117-t003:** Summary of univariate analysis of poultry samples for antimicrobial-resistant *E. coli* from poultry farms in east coast of Malaysia (n = 371 samples).

Antimicrobials	Cloacal n = 259	Faecal n = 84	Sewage n = 14	Tape Water n = 14	*p*-Value
No identified resistance					<0.001
No antimicrobial resistance	89 (34.4%)	26 (31%)	9 (64.3%)	13 (92.9%)	
Resistance to at least one antimicrobial	170 (65.6%)	58 (69%)	5 (35.7%)	1 (7.1%)	
Antimicrobial class resistance					0.003
No antimicrobial resistance	89 (34.4%)	26 (31%)	9 (64.3%)	13 (92.9%)	
Resistant to 1 class	4 (1.5%)	1 (1.2%)	0 (0%)	0 (0%)	
Resistant to 2 classes	13 (5%)	3 (3.6%)	2 (14.3%)	0 (0%)	
Resistant to 3 classes	46 (17.8%)	19 (22.6%)	2 (14.3%)	1 (7.1%)	
Resistant to 4 classes	74 (28.6%)	19 (22.6%)	1 (7.1%)	0 (0%)	
Resistant to 5 or more classes	33 (12.7%)	16 (19%)	0 (0%)	0 (0%)	
Source of antimicrobials					1
Drug supplier	112 (43.2%)	36 (42.9%)	6 (42.9%)	6 (42.9%)	
Feed store	147 (56.8%)	48 (57.1%)	8 (57.1%)	8 (57.1%)	
Tetracyclines					<0.001
Not resistant	93 (35.9%)	26 (31%)	9 (64.3%)	13 (92.9%)	
Resistant	166 (64.1%)	58 (69%)	5 (35.7%)	1 (7.1%)	
Penicillins					0.048
Not resistant	151 (58.3%)	47 (56%)	10 (71.4%)	13 (92.9%)	
Resistant	108 (41.7%)	37 (44%)	4 (28.6%)	1 (7.1%)	
Aminoglycosides					0.246
Not resistant	219 (84.6%)	68 (81%)	13 (92.9%)	14 (100%)	
Resistant	40 (15.4%)	16 (19%)	1 (7.1%)	0 (0%)	
Quinolones					0.002
Not resistant	142 (54.8%)	49 (58.3%)	13 (92.9%)	13 (92.9%)	
Resistant	117 (45.2%)	35 (41.7%)	1 (7.1%)	1 (7.1%)	
Sulfonamides					<0.001
Not resistant	104 (40.2%)	29 (34.5%)	11 (78.6%)	14 (100%)	
Resistant	155 (59.8%)	55 (65.5%)	3 (21.4%)	0 (0%)	
Cephelosporins					0.645
Not resistant	246 (95%)	79 (94%)	14 (100%)	14 (100%)	
Resistant	13 (5%)	5 (6%)	0 (0%)	0 (0%)	
Other classes					0.025
Not resistant	224 (86.5%)	65 (77.4%)	14 (100%)	14 (100%)	
Resistant	35 (13.5%)	19 (22.6%)	0 (0%)	0 (0%)	

**Table 4 antibiotics-10-00117-t004:** Univariate regression analysis of risk factors for antimicrobial-resistant *E. coli* from poultry farms in east coast of Malaysia (n = 238 samples).

Variables	OR	2.5%	97.5%	Pr (>|z|)	
Farms					
Farm 1	10.20	2.95	42.89	<0.001	***
Farm 2	2.72	0.91	8.53	0.07	.
Farm 3	13.03	3.46	65.56	<0.001	***
Farm 4	1.98	0.66	6.09	0.22	
Farm 5	2.31	0.78	7.18	0.134	
Farm 6	7.14	2.16	27.16	0.002	**
Farm 7	2.89	0.97	9.02	0.05	.
Farm 8	Ref	Ref	Ref	Ref	
Farm 9	4.61	1.48	15.63	0.01	*
Farm 10	3.82	1.25	12.52	0.02	*
Farm 11	1.16	0.38	3.53	0.78	
Farm 12	1.06	0.34	3.25	0.91	
Farm 13	5.66	1.78	20.13	0.004	**
Farm 14	2.26	0.77	6.87	0.13	
Sample source					
Cloaca swab	24.83	4.82	454.74	0.002	**
Faecal sample	29.0	5.35	540.73	0.001	**
Sewage	7.22	0.95	151.21	0.09	.
Tap water	Ref	Ref	Ref	Ref	
Age					
Young	1.21	0.79	1.84	0.38	
Adult	Ref	Ref	Ref	Ref	
Poultry origin					
Local	1.44	0.61	3.50	0.41	
Both	2.08	1.32	3.26	0.001	**
Imported	Ref	Ref	Ref	Ref	
Management system					
Semi-intensive	1.52	0.97	2.39	0.06	.
Mixed	1.11	0.48	2.65	0.80	
Intensive	Ref	Ref	Ref	Ref	
Production system					
Layer	1.41	0.76	2.69	0.27	
Broiler	Ref	Ref	Ref	0.11	
Mixed	1.50	0.91	2.48	Ref	
Farm size					
Small	2.33	1.27	4.35	0.001	**
Medium	1.51	0.88	2.57	0.125	
Large	Ref	Ref	Ref	Ref	
Water source					
Surface water	1.57	0.93	2.68	0.08	.
Pump water	2.02	1.22	3.36	0.01	**
Bond water					
Sewage system					
Excellent	0.91	0.44	1.81	0.786	
Good	0.75	0.39	1.42	0.398	
Poor	Ref	Ref	Ref	Ref	

Signif. codes: 0 ‘***’ 0.001 ‘**’ 0.01 ‘*’ 0.05 ‘.’ 0.1 ‘ ‘ 1.

**Table 5 antibiotics-10-00117-t005:** Multivariate regression analysis of risk factors for antimicrobial-resistant *E. coli* from poultry farms in east coast of Malaysia.

	OR	2.5%	97.5%	Pr (>|z|)	
Cloaca swab	26.50	5.08	487.69	0.001	**
Feacal sample	30.92	5.63	579.63	0.001	**
Sewage	7.43	0.96	156.87	0.09	.
Farm size Small	2.50	1.33	4.77	0.004	**
Farm size medium	1.55	0.89	2.67	0.114	

Signif. codes: 0 ‘***’ 0.001 ‘**’ 0.01 ‘*’ 0.05 ‘.’ 0.1 ‘ ‘ 1.

**Table 6 antibiotics-10-00117-t006:** Comparison for the detection of resistance genes from samples using PCR.

Antimicrobial Class/Agent	Resistance Gene	% Isolates	Total # Tested
Gentamicin	aac(3)-IV	12 (85.7%)	14
Tetracyclines	tet(A), tet(B)	9 (64.2%)	14
Chloramphenicol	catA1	2 (14.2%)	14
Sulfonamides	sul1	14 (100%)	14
β-Lactams	blaSHV	6 (42.8%)	14
Trimethoprim	dhfrI	4 (28.5%)	14

**Table 7 antibiotics-10-00117-t007:** The set of primers used for each gene.

Genes	Primer Sequence(5′ to 3′)	PCR Condition	Product Size	References
β-Lactams	F- CTATCGCCAGCAGGATCTGG R- ATTTGCTGATTTCGCTCGGC	3 min at 95 °C; 35 cycles of 1 min at 94 °C, 90 s at 55 °C and 1 min at 72 °C; 10 min at 72 °C	543	[16]
Gentamicin *aac(3)-IV*	F-CTTCAGGATGGCAAGTTGGT R-TCATCTCGTTCTCCGCTCAT	3 min at 95 °C; 35 cycles of 1 min at 94 °C, 90 s at 55 °C and 1 min at 72 °C; 10 min at 72 °C	286	[17]
Sulfonamide *sul1*	F- ACTGCAGGCTGGTGGTTATG R- ACCGAGACCAATAGCGGAAG	3 min at 95 C; 35 cycles of 1 min at 94 C, 90 s at 55 °C and 1 min at 72 °C; 10 min at 72 °C	271	[8]
Tetracycline *tet(A)*	F-CCTCAATTTCCTGACGGGCT R-GGCAGAGCAGGGAAAGGAAT	3 min at 95 °C; 35 cycles of 1 min at 94 C, 90 s at 55 °C and 1 min at 72 °C; 10 min at 72 °C	712	[18]
Tetracycline *tet(B)*	F-ACCACCTCAGCTTCTCAACG R-GTAAAGCGATCCCACCACCA	3 min at 95 °C; 35 cycles of 1 min at 94 C, 90 s at 55 °C and 1 min at 72 °C; 10 min at 72 °C	586	[18]
Chloramphenicol *catA1*	F- GAAAGACGGTGAGCTGGTGA R- TAGCACCAGGCGTTTAAGGG	3 min at 95 °C; 35 cycles of 1 min at 94 °C, 90 s at 55 °C and 1 min at 72 °C; 10 min at 72 °C	473	[8]
Trimethoprim dhfrI	F-AAGAATGGAGTTATCGGGAATG R-GGGTAAAAACTGGCCTAAAATTG	15 min at 95 °C; 30 cycles of 30 s at 94 °C; 30 s at 58 °C; 1 min at 72 °C; 10 min 72 °C.	391	[8]
Ampicillin *CITM*	F-TGGCCAGAACTGACAGGCAAA R-TTTCTCCTGAACGTGGCTGGC	15 min at 95 °C; 30 cycles of 30 s at 94 °C; 30 s at 58 °C; 1 min at 72 °C; 10 min 72 °C.	462	[8]
*E.coli*	F-TGACGTTACCCGCAGAAGAA R- CTCCAATCCGGACTACGACG	3 min at 95 °C; 35 cycles of 15s at 95 °C, 90 s at 55 °C and 15s at 72 °C; 10 min at 72 °C	832	[19]
O157	F-GTGTCCATTTATACGGACATCCATG R-CCTATAACGTCATGCCAATATTGCC	2 min at 94 °C; 35 cycles of 30s at 94 °C, 30 s at 55 °C and 30s at 72 °C; 5 min at 72 °C	292	[20]

## Data Availability

The data for the present study will be available upon reasonable request.

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
