# Peer review of "Identification of Risk Factors Associated with Resistant Escherichia coli Isolates from Poultry Farms in the East Coast of Peninsular Malaysia: A Cross Sectional Study"

_antibiotics, 2021, doi:10.3390/antibiotics10020117_

Round 1

Reviewer 1 Report

The manuscript authored by Sharifa et al. attempted to describe an overall prevalence and risk factors of antimicrobial resistance in Escherichia coli isolated from some regional poultry farms in Malaysia. The overall objectives of the study are very important in the field of antimicrobial resistance. However, the methodology, results and discussions are not well explained and there are data discrepancies across different sections. Following are some of my observations and specific comments.

Data discrepancies

Line 31 and line 91: What are the actual number of strains isolated in this study? In the abstract, it is mentioned that 717 E. coli strains were isolated (Line 31) but the result section shows 1,113 isolates (line 91).

Line 33: The odds ratio for surface water are shown to be 1.55 here. This number is never found in the respective table (Line 188).

Line 33-34: What is the meaning of P value less than 0.000? And these P values are not shown inside the manuscript or table.

Methodology

The “microbiological testing” section needs improvement.

Line 324 and line 329: “five colonies were selected (line 324)” then what does it mean by “A single colony was picked from each individual animal”.

Line 326: Is there any reason for preserving the isolate at -20? As the standard cryopreservation temperature is -80.

Results

Line 94: “The prevalence of E. coli sapmles ranges from 4.6 % to 10.1% across all 14 farms”. It is difficult to understand where is this prevalence come from? Is it from cloacal swab or other source? And the Table 1 also lacking sample types. If it is cloacal swab samples then the prevalence is way underestimated, I guess. E. coli is a very prevalent commensal gut bacterium in warm blooded animals including chicken. So naturally the chicken cloacal swab should contain more E. coli. Is there any reasoning behind this?

Line 111-113: This doesn’t make any sense. “The poultry farms practicing intensive management system were more likely to carry antimicrobial-resistant E. coli than those practicing intensive system”.

Line 109, 121-128: The author started describing water source as a risk factor for AMR acquisition in line 109. Then they began describing other risk factors in that paragraph. Again, they opened a new paragraph and began talking about water source as a risk factor (line 121). And then they started describing PCR results in the same paragraph (Line 122-128). It seems the author didn’t have any organized plan to demonstrate the results.

Discussion

Line 236-252: The authors tried to justify their findings with some unpublished findings and facts. Some of the justification is out of the context and difficult to understand. For instance, in line 239-241, “This indicates that the water system itself may be contaminated with either bacterial colonies of contain high levels of environmental antibiotics and needs further investigation in these poultry operations”. I can’t corelate this with the risk factors that was detected in this study.

Line 244-245: “The discovery of these antibiotic contents could be most likely attributed to potential contaminations from livestock farming discharge field runoff.” What is this discovery? Nothing is mentioned in the results.

Line 253-269: I could not follow this part of the discussion.

Others

The authors used both term “antimicrobials” and “antibiotic” synonymously. However, these two terms are not synonymous. I would suggest using “antimicrobials” across the entire manuscript.

Author Response

POINT BY POINT RESPONSE TO THE REVIEWERS SUGGESTION

RESPONSE TO REVIEWER 1:

We thank the reviewer for reviewing this paper and for the comments and suggestions.  We have tried at our best to revise the manuscript in line with these comments and suggestions. We would also like share with the reviewer that we have re-analyzed the whole data based on samples tested in this manuscript.

The specific changes and response to the different points raised include: 

Data discrepancies

  1. Line 31 and line 91: What are the actual number of strains isolated in this study? In the abstract, it is mentioned that 717 coli strains were isolated (Line 31) but the result section shows 1,113 isolates (line 91).

Response: Thank you for your valuable comments. The actual number of strains isolated is 717 E. coli strains NOT the [1, 113] which refers to the triplicate sub-culturing of the original samples (371) (see line 98). 

  1. Line 33: The odds ratio for surface water are shown to be 1.55 here. This number is never found in the respective table (Line 188).

Response: This is typos and we have rectified 1.57 instead of 1.55. The OR and the CI remains correct (see line  264).

  1. Line 33-34: What is the meaning of P value less than 0.000? And these P values are not shown inside the manuscript or table.

Response:  Now, we have updated the P values less than 0.01 in line 33-34. Furthermore, we added additional column of the P values of each variable in the table 4 (see line 194-197).

Methodology

  1. Line 324 and line 329: “five colonies were selected (line 324)” then what does it mean by “A single colony was picked from each individual animal”.

Response: We have deleted ‘A single colony was picked from each individual animal”.’ The single colony was ONLY selected from each original sample (see line 383-384).

  1. Line 326: Is there any reason for preserving the isolate at -20? As the standard cryopreservation temperature is -80.

Response: We agree the referee’s assertion, but we were not in favor to preserve these isolates for long time or to revive them in the future along with space constraints in the facility.  Moreover, we have also run the revival of these isolates within short period of time and thus we have sufficiently satisfied with -20 which seemed appropriate according to our circumstances.  

Results

  1. Line 94: “The prevalence of coli samples ranges from 4.6 % to 10.1% across all 14 farms”. It is difficult to understand where is this prevalence come from? Is it from cloacal swab or other source? And the Table 1 also lacking sample types. If it is cloacal swab samples then the prevalence is way underestimated, I guess. E. coli is a very prevalent commensal gut bacterium in warm blooded animals including chicken. So naturally the chicken cloacal swab should contain more E. coli. Is there any reasoning behind this?

Response: Thank you for your comment. We have omitted the ‘The prevalence of E. coli samples ranges from 4.6 % to 10.1% across all 14 farms’ to avoid confusion and thus we have replaced the original table with un updated version in which we clearly calculated the prevalence of each variable in more meaningful way (see line 171-175). 

  1. Line 111-113: This doesn’t make any sense. “The poultry farms practicing intensive management system were more likely to carry antimicrobial-resistant coli than those practicing intensive system”.

Response: We have deleted this statement from the revised version of the manuscript.

  1. Line 109, 121-128: The author started describing water source as a risk factor for AMR acquisition in line 109. Then, they began describing other risk factors in that paragraph. Again, they opened a new paragraph and began talking about water source as a risk factor (line 121). And then they started describing PCR results in the same paragraph (Line 122-128). It seems the author didn’t have any organized plan to demonstrate the results.

Response: We have done our best to improve the discussions. (See line 441-473)

Discussion

  1. Line 236-252: The authors tried to justify their findings with some unpublished findings and facts. Some of the justification is out of the context and difficult to understand. For instance, in line 239-241, “This indicates that the water system itself may be contaminated with either bacterial colonies of contain high levels of environmental antibiotics and needs further investigation in these poultry operations”. I can’t corelate this with the risk factors that was detected in this study.

Response: Thank you for your comment. This is a large study in which we have collected poultry and environmental samples including rural water systems which in close proximity to the farms and we have screened and quantified the residuals of veterinary antibiotics in these tributaries using LCMS/MS-Q-TOF and lCMS-MS-QQQ, respectively along with susceptibility of E. coli and salmonella in collaboration with Monash University, Malaysia. We have tried to justify these results with our unpublished findings to make sense out of it. Moreover, we have excluded  the sentence ‘This indicates that the water system itself may be contaminated with either bacterial colonies of contain high levels of environmental antibiotics and needs further investigation in these poultry operations” from the discussion (see line 264-266 .

  1. Line 244-245: “The discovery of these antibiotic contents could be most likely attributed to potential contaminations from livestock farming discharge field runoff.” What is this discovery? Nothing is mentioned in the results.

Response: More information regarding the unpublished finings has now been added. This has been detailed in the manuscript as follows: ‘Most of these antimicrobial residues belong to Sulfonamides and Quinolones in the surface water at an average concentration range of 5 to 85 ng L-1. Conversely, we have detected low levels of Tetracyclines in the surface water, although higher levels of TCs were detected in the faecal samples of poultry operations and in the Kelantanese tributaries sediments.’ see line 267-272 .

  1. Line 253-269: I could not follow this part of the discussion.

Response: Despite, the short deadline, we feel that the paper is much improved as a result of this peer review process, as we have also re-analyzed the data.

Others

  1. The authors used both term “antimicrobials” and “antibiotic” synonymously. However, these two terms are not synonymous. I would suggest using “antimicrobials” across the entire manuscript.

Response: Thank you for your valuable suggestions. We have now used “antimicrobials” across the entire manuscript.

While hoping that these amendments would meet with your favorable consideration, we meanwhile remain completely open to any further suggestions.

***

Reviewer 2 Report

Antibiotic resistance is a growing global menace. The poultry makes up a substantial portion of the global antimicrobial uses. In this manuscript, Osman and coworkers describe the prevalence and risk factors associated with E.coli isolated from poultry farms on peninsular Malaysia's east coast. It cannot be accepted in its current form due to it experiment design and data analysis. Authors are suggested to address the following items. 

  1. The authors should explain the error bars in Figure 2. The authors did not conduct any replication in antimicrobial susceptibility testing. Also, the data in this figure does not support the authors’ conclusion in Line 102 and 103.
  2. Table 5: It makes no sense to separate ampicillin from β-lactams since ampicillin belongs to β-lactams. Also, CITM is NOT the best choice for this study. CITM is one of the AmpC β-lactamases, which are clinically important cephalosporinases.
  3. Line 94: Authors claim that “the prevalence of E.coli samples ranges from 4.6 to 10.1 across all 14 farms”. It is hard for readers to understand the sentence without knowing the way to calculate the % positive in Table 1. 
  4. Table 2: authors need to provide the definitions or criteria for the management system, farm size, and sewage system. For example, what is a poor sewage system? Besides, the author should change “fecal sample” to “ Faecal samples” in this table. 
  5. Line 322: Authors claimed that “sewage, tap water and surface water samples were transported”. However, it seems that surface water samples were never collected for this study. A total of 371 samples only include cloacal swabs (259), faecal (84), sewage (14) and tap water (14) (see line 306). The author should clarify it.
  6. Line 297: add space between “China Sea” and “and”.
  7. Line 94: authors need to change “sapmles” to “samples.”
  8. Line 104: authors need to correct the word “summarized.” 

Author Response

RESPONSE TO REVIEWER 2:

We thank the reviewer for reviewing this paper and for the comments and suggestions.  We have tried at our best to revise the manuscript in line with these comments and suggestions. We would also like share with the reviewer that we have re-analyzed the whole data based on samples tested in this manuscript.

The specific changes and response to the different points raised include: 

  1. The authors should explain the error bars in Figure 2. The authors did not conduct any replication in antimicrobial susceptibility testing. Also, the data in this figure does not support the authors’ conclusion in Line 102 and 103.

Response: We agree with the reviewer and thus deleted the error bars in figure 1 and 2 along with the conclusion statement ‘Moreover, the prevalence of these resistances was high in Kelantan and Pahang poultry farms and low in Terengganu (Figure 2).’ in line (157 and 156).

  1. Table 5: It makes no sense to separate ampicillin from β-lactams since ampicillin belongs to β-lactams. Also, CITM is NOT the best choice for this study. CITM is one of the AmpC β-lactamases, which are clinically important cephalosporinases.

Response: Thank you for your comments. We have no deleted the ampicillin column from table 5 (now table 6) section (see line 206-2014).

  1. Line 94: Authors claim that “the prevalence of E.coli samples ranges from 4.6 to 10.1 across all 14 farms”. It is hard for readers to understand the sentence without knowing the way to calculate the % positive in Table 1.

Response: Thank you for your valuable comments. We agree with the reviewer and thus we have re-analyzed the data using Chis square analysis to calculate the prevalence in more manful way so that the readers can easily understand it (kindly see Table 1; line 171-175).

  1. Table 2: authors need to provide the definitions or criteria for the management system, farm size, and sewage system. For example, what is a poor sewage system? Besides, the author should change “fecal sample” to “ Faecal samples” in this table.

Response: We did our best to thoroughly provide the definitions for the management, farm size and sewage system.  This has been detailed in the manuscript (see line 346-368) as follows:

Regarding, the management system, flock size and sewage system, the following definitions and criteria were used:

  • intensive management system is defined as mainly concentrated and often mechanized operations that use controlled-environment systems to provide the ideal thermal environment for the poultry.
  • Semi-intensive system is that which relies on natural airflow though the shed for ventilation.
  • Extensive system is mainly a pasture-based and land-based where birds in the household flock are typically housed overnight in the shelter and are let out in the morning to forage during the day.
  • The criteria of the farm size included large-scale commercial farms that has more than ≥10,000 birds; Medium-scale commercial farms that has more 5,000- 10,000 and small-scale farms where birds are often kept in single-age groups of >1000.
  • Poor sewage system is defined that which retains high volumes of wastewater with low flow rate, blackish appearance and sewage smell odour as results of composing agricultural waste –probably as leakage from nearby irrigated effluent which is used for agricultural land application along with the presence of food waste, Green waste, plastic and heavy materials.
  • Good sewage system is that which has a good drainage with no agricultural waste and relatively low heavy materials.
  • Excellent swage systems is that which has significant drainage, no agricultural and heavy materials.

***Of Note, We have only ranked the sewage system based on physical appearance and it has to be emphasized that this ranking system is only relevant to our situation, since they are strongly affected by case and time specificity.

  1. Line 322: Authors claimed that “sewage, tap water and surface water samples were transported”. However, it seems that surface water samples were never collected for this study. A total of 371 samples only include cloacal swabs (259), faecal (84), sewage (14) and tap water (14) (see line 306). The author should clarify it.

Response: We agree with the reviewer that no surface water samples were included. Although, this is quite large study in which we have also collected survey water from some parts of Kelantanese tributaries. Unfortunately, the lab work of the surface water and the detection of antibiotic residues remains yet incomplete and requires further process and data analysis. Therefore, we refrained the preliminary work of the surface water from the current manuscript.  However, it’s worth noting, that preliminary results (unpublished) detected antibiotics residue that belong to Sulfonamides and Quinolones in the surface water at an average concentration range of 5 to 85 ng L-1.

  1. Line 297: add space between “China Sea” and “and”.

Response: revised as suggested.

  1. Line 94: authors need to change “sapmles” to “samples.”

Response: revised as suggested.

  1. Line 104: authors need to correct the word “summarized.

Response: revised as suggested.

While hoping that these amendments would meet with your favorable consideration, we meanwhile remain completely open to any further suggestions.

***

Round 2

Reviewer 1 Report

The authors resolved all my comments. I don't have any further queries. 

Reviewer 2 Report

The authors have addressed my concerns.